# A Quest for Survival: A Review of the Early Biomarkers of Pancreatic Cancer and the Most Effective Approaches at Present

**DOI:** 10.3390/biom14030364

**Published:** 2024-03-19

**Authors:** Muhammad Begawan Bestari, Ignatius Ronaldi Joewono, Ari Fahrial Syam

**Affiliations:** 1Division of Gastroenterohepatology, Department of Internal Medicine, Hasan Sadikin General Hospital, Faculty of Medicine, University of Padjadjaran, Bandung 40161, Indonesia; 2Mochtar Riady Comprehensive Cancer Center, Siloam Hospitals, Jakarta 12930, Indonesia; ignatius.joewono@gmail.com; 3Division of Gastroenterology, Department of Internal Medicine, Cipto Mangunkusumo National Central Hospital, Faculty of Medicine, University of Indonesia, Jakarta 10430, Indonesia; ari_syam@hotmail.com

**Keywords:** pancreatic cancer, early biomarker, early diagnosis

## Abstract

Pancreatic cancer (PC) is the most lethal type of cancer; it has the lowest 5-year survival rate among all other types of cancers. More than half of PC cases are diagnosed at an advanced stage due to PC’s insidious and non-specific symptoms. Surgery remains the most efficacious treatment option currently available, but only 10–20% of PC cases are resectable upon diagnosis. As of now, the sole biomarker approved by the United States Food and Drug Administration (US-FDA) for PC is carbohydrate antigen 19-9 (CA19-9); however, its use is limited for early diagnosis. An increasing number of studies have investigated a combination of biomarkers. Lately, there has been considerable interest in the application of a liquid biopsy, including the utilization of *microRNAs* (*miRNAs*), *circulating tumor DNA* (*ctDNA*), and circulating tumor cells (CTCs). Screening for PC is indicated for high-risk patients; studies on new diagnostic models combined with biomarkers for early detection have also shown promising results in terms of the ability of these models and biomarkers to aid clinicians in deciding on whether to start screening. This review seeks to provide a concise overview of the advancements in relation to existing biomarkers and explore novel strategies for the early detection of PC.

## 1. Introduction

Pancreatic cancer (PC) is currently the most lethal type of cancer, exhibiting the lowest 5-year survival rate among all cancer types. The survival rate is estimated at 8% for all PC stages combined and only 3% when PC is diagnosed at a later stage [1]. More than 50% of PC patients are diagnosed at an advanced stage, at which point there are only limited treatment options due to the highly metastatic nature of this disease [1,2,3]. According to GLOBOCAN, global statistics indicate that PC leads to approximately 432,000 deaths, with 459,000 new cases reported in 2018 [4]. In contrast to other cancer types with declining rates, PC is expected to rank as the third most common cause of cancer-related fatalities by the year 2025, surpassing even breast cancer [5].

At an early stage, PC rarely shows any symptoms. If there are any symptoms at all, they are mostly non-specific. The common symptoms include weight loss, abdominal pain, steatorrhea, and new-onset diabetes or the deterioration of pre-existing diabetes. In the advance stage, symptoms of jaundice and light-colored stools, as well as other symptoms caused by the obstruction of the common biliary duct and/or pancreatic duct, will start to appear [6]. It is noteworthy that PC is frequently discovered in autopsy studies due to its subtle and insidious nature [7,8]. Approximately 63% of PCs originate in the head of the pancreas, with those originating in the tail and body accounting for around 12.8% and 9.8% of cases, respectively [3]. Tumors originating in the tail and body tend to be detected at more advanced stages, as they require more time to develop noticeable symptoms.

Imaging studies also play a role in the detection of PC. They are used to assess the tumor location and size, the vascular involvement, the regional involvement, and the metastatic extent (e.g., liver, lungs, and peritoneum) [6]. Ultrasound lacks the specificity and sensitivity required for the detection of small PC lesions due to the presence of gas in the gastrointestinal tract, making an early diagnosis difficult [9]. A computed tomography (CT) scan with contrast remains the main modality used to diagnose PC; for cancers smaller than 2 cm, it has a sensitivity of up to 63–77% [10]. In cases where a CT scan yields inconclusive results, magnetic resonance imaging (MRI) can be employed, particularly for isoattenuating PC [11]. However, neither CT scans nor MRI guarantee tumor detection, especially when jaundice is already present. Consequently, an endoscopic ultrasound (EUS) examination has becomes valuable for diagnosis [12]. An EUS has demonstrated diagnostic rates of 45.5% for stage 0 and 81.8% for stage I cancers, surpassing the rates of 9.7% and 63% for CT scans, as well as 9.7% and 39.1% for MRI, respectively [13,14].

A PC patient’s prognosis is primarily determined by utilizing a tumor node metastasis (TNM) staging system [6]. The American Joint Committee on Cancer (AJCC) classifies the TNM system for PC into four stages: stage I—a localized, resectable tumor, smaller than 2 cm (IA) or larger than 2 cm but smaller than 4 cm (IB); stage II—a larger tumor (>4 cm) limited to the pancreas (IIA) or involving 1–3 regional lymph nodes (LNs) with a tumor size <4 cm (IIB); stage III—metastasis to ≥4 regional LNs, regardless of the tumor size; and stage IV—distant metastasis [15]. The two most important prognostic factors are the size of the tumor upon detection (<2 cm) and detection at an early stage [16]. Data from the United States National Cancer Institute reveal that only 12% of PC cases are detected at the local stage, which boasts a 5-year survival rate of 44%. The 5-year survival rate drastically drops to 15% when the cancer is detected at stage III, which involves the surrounding tissue, and for distant metastasis, the point at which more than half of all patients are detected, the survival rate is only 3% [17]. Surgery remains the most effective treatment for PC when diagnosed early; however, only 10–20% of cases are eligible for surgical resection upon detection [18,19]. In patients diagnosed with metastatic tumors, the median survival time is only 3 months, and it is 6–9 months for locally advanced stage cancer [20]. There is an urgent need to find diagnostic tools and methods that can detect PC at the earliest stage; our review will highlight novel biomarkers and the feasibility of using them.

Developing pancreatic screening criteria is challenging, and population-based screening is not feasible. In the general population, with no risk stratification applied, the approximate lifetime risk of developing PC up to the age of 70 years is only 1.3% [20]. Screening the general population is not cost-effective, and there is no supporting evidence indicating a reduction in mortality [21,22]. Likewise, the US Preventive Service Task Force advises against screening asymptomatic patients [21,23,24].

Routine screening is recommended for individuals with inherited genetic abnormalities, such as a familial history of PC or Peutz–Jeghers syndrome [25]. Researchers are currently engaged in an ongoing debate regarding the ideal age at which to commence the initial screening process. Generally, it is suggested that screening should commence around the age of 40–50 years or 10–15 years earlier than the age of onset observed in family members diagnosed with PC [26]. The International Cancer of the Pancreas Screening Consortium recommends that surveillance should begin at 50 years old or 10 years earlier than the youngest age that a blood relative with PC was diagnosed. Screening is performed every 3 years, or every 3–6 months if there are abnormalities [26,27].

The NCCN recommends an EUS as a screening tool. In a study of 78 high-risk individuals, CT and an EUS successfully identified 8 patients with PC, 6 patients with an intraductal papillary mucinous neoplasm (IPMN), and 3 patients with an extra pancreatic neoplasm [28,29,30]. Canto et al. reported that an EUS, MRI, and CT could detect pancreatic lesions in asymptomatic patients, with the detection rates reaching 42.6%, 33.3%, and 11.0%, respectively [25]. According to the International Cancer of the Pancreas Screening Consortium, three-fourths of experts agree that an EUS and MRI are the preferred screening methods over CT, which is attributed to their higher detection rates. However, there is no consensus on the optimal frequency for conducting screening [23].

Several unhealthy lifestyles and living habits have been linked to an increased risk of developing PC. Cigarette smoking has been documented to elevate the risk of PC by two to three times [31]. A heavy alcohol intake has been described as an independent risk factor that contributes to the PC risk in men (hazard ratio (HR) = 1.69, 95% CI: 1.21–2.37) [32,33]. Another significant risk factor is diabetes mellitus (DM). An association between DM and PC has been observed since the 1800s; however, the exact mechanism is still not fully understood [34,35]. The prevalence of DM in PC patients ranges from 4 to 65% [36,37,38]. In a study conducted by Pannala et al., 47% of the PC cases were identified in individuals with diabetes mellitus (DM), as opposed to the control group, where only 7% had DM. Notably, 74% of those with DM in the PC group had new-onset diabetes [39]. A disturbance in glucose homeostasis is universally observed in PC patients and pancreatic ductal adenocarcinoma (PDAC) is recognized as the most consistent diabetogenic cause in humans, making it the most common phenotypic trait of PC [40]. Abnormalities in fasting blood glucose levels have been detected 30–36 months before a PC diagnosis, progressively increasing up to 126 mg/dL approximately 6–12 months before the cancer diagnosis [41]. Another study also reported that the mean interval between the onset of diabetes and PC is 10 months, ranging from 5 to 29 months [42]. A clinical diagnostic model known as the enriching new-onset diabetes for PC (ENDPAC) model, which utilizes three parameters (age, changes in blood glucose, and changes in body weight), successfully identified patients who developed PC within three years of diabetes onset, with an AUC of 0.87, a sensitivity of 80%, and a specificity of 80% [43]. Another clinical model established by the Peking Union Medical College Hospital (PUMCH), which incorporates 10 risk factors (gender, age, alcoholic intake, smoking, diabetes mellitus history, high meat consumption, a family history of PC, chronic pancreatitis, cholelithiasis history, and cholecystitis history), achieved a sensitivity of 88.9%, a specificity of 97.6%, and an AUC of 0.98 [44].

## 2. Current Biomarkers

The only biomarker currently accepted by the FDA and the National Comprehensive Cancer Network guidelines for PC is carbohydrate antigen 19-9 (CA19-9) [28,29]. Although CA19-9 can serve as a prognostic factor, its utility for early diagnosis and screening is limited [45,46]. The ESMO clinical guidelines for PC explicitly state that CA19-9 is not effective as a screening tool [6]. Previous studies have stated that CA19-9 has a mean sensitivity of 78.2% and a specificity of 82.8% for identifying pancreatic carcinoma [47]. Elevated CA19-9 levels are also observed in various other tumor types, including colorectal cancer, cholangiocarcinoma, liver cancer, and gastric cancer. Furthermore, in some benign conditions, such as obstructive jaundice and cirrhosis, this tumor marker also shows elevated levels [14,48]. Because CA19-9 is a sialylated Lewis antigen blood group, people in the Lewis-antigen-negative blood group will not synthesize it. It has been estimated that up to 10% of people do not express Lewis antigens [49,50]. Notably, a recent study by Liu et al. reported that individuals lacking Lewis antigens experienced poorer outcomes when diagnosed with PC, including higher metastatic rates [51].

The second most common biomarker used for PC is the carcinoembryonic antigen (CEA). Its reported sensitivity and specificity are 44.2% and 84.8%, respectively [47]. Similar to CA19-9, the CEA is predominantly utilized as a prognostic marker of PC. Elevated CEA levels have been observed in 30–60% of PC patients [52]. The CEA is also found in other types of cancers, including those affecting the colon, breast, lung, and thyroid. When used as a solitary biomarker, the CEA exhibits a sensitivity of 43% and a specificity of 82%, which are even lower than those of CA19-9 [53]. Moreover, the CEA can be detected in non-cancer diseases, such as in cigarette smokers, people with cholecystitis, liver cirrhosis, pancreatitis, inflammatory bowel disease, or COVID-19, and in people using medications such as orlistat [54,55].

CA19-9 and the CEA can also serve as tools for patient stratification during diagnosis. CA19-9 can act as a predictor of lymph node metastasis, with a cut-off value of 400 U/mL [56]. A recent study conducted by Esen et al. indicated that CA19-9 alone is effective in distinguishing N0 from N2 patients, although it cannot identify N1 patients [57]. However, it has been noted that utilizing a CA19-9/CEA ratio can differentiate whether a patient is an N0, N1, or N2 patient. The reported sensitivity and specificity of the CA19-9/CEA ratio, with a cut-off value of 27.18, are 79.4% and 80.4%, respectively [57]. The capability of differentiating N0 from N1 tumors would be a valuable tool, given that N0 patients have a higher likelihood of undergoing complete surgical resection.

## 3. Novel Biomarkers

### 3.1. Proteomic Biomarkers

Emerging protein biomarkers have demonstrated the potential to detect early-stage PC. Leucine-rich alpha-2 glycoprotein 1 (LRG1) is a glycoprotein that is part of the leucine-rich repeat (LRR) family of proteins. Its primary functions include an involvement in protein interactions, signal transduction, and cell adhesion and development, as well as the facilitation of new blood vessel formation. An elevated expression of LRG1 has been associated with a poor survival and an advanced tumor stage. Moreover, LRG1 has been implicated in promoting the viability, proliferation, and invasion of pancreatic tumor cells [58,59,60,61,62]. Matrix metalloproteinases (MMPs) form a group of proteases recognized for their capability to degrade extracellular matrix components, including gelatinase B (MMP-9), which is acknowledged for digesting the primary constituent of the basement membrane (type IV collagen). The degradation of the extracellular matrix and basement membranes is pivotal in cancer invasion and metastasis, indicating that changes in the matrix metalloproteinase (MMP) activity within the tumor environment likely play a role in the progression of PC. However, despite its role, circulating MMP-9 has been reported as an inferior marker of PC when compared to CA19-9. Even when both markers are combined, the diagnostic accuracy does not improve [63]. The clinical relevance of MMP-9 concerning the survival, metastasis, and tumor stage has been observed diversely in various studies [64]. Tissue inhibitors of metalloproteinases (TIMPs) belong to another class of metalloproteinases capable of binding to MMPs and thereby exerting inhibitory and activating effects on MMPs and potentially being involved in tumor progression [63]. TIMP-1, which is typically expressed to regulate cell proliferation and apoptosis, has been identified as a potential biomarker for a PC diagnosis, with a sensitivity of 47.1%, a specificity of 69.2%, and an AUC of 0.64 [63,65]. Transthyretin (TTR), a carrier of thyroid hormones (thyroxin and tri-iodothyronine), has been found to increase by more than 1.5-fold in the serum of PDAC patients compared to normal controls. This increase is associated with a sensitivity of 90.5%, a specificity of 47.6%, and an AUC of 0.75 [66]. Intercellular adhesion molecule 1 (ICAM-1), a glycoprotein that plays a role in cell adhesion and acts as a macrophage chemoattractant, has been assessed in multiple studies as a potential early diagnostic tool for PC. When using a cut-off value of 878.5 U/mL, ICAM-1 exhibited a sensitivity, specificity, and AUC of 82%, 82.26%, and 0.851, respectively [67]. Osteoprotegerin (OPG), known for its role in bone homeostasis, has emerged as a potential biomarker for the early detection of PC. Shi et al. reported that OPG is upregulated in cancerous pancreatic tissue, with an even higher expression observed in patients experiencing new-onset diabetes [68,69,70].

Chemokines, also known as chemotactic cytokines, constitute a group of proteins that regulate the migration, adhesion, growth, activation, and differentiation of leukocytes. They are categorized into four groups based on the key cysteine positions: CC, CXC, CX3C, and XC [71,72]. Chemokines play a pivotal role in the modulation of inflammation, infection, immune responses, tissue injury, and various pathological processes, including the development of malignancies [73,74]. The expression of the CXCL-1 chemokine in PC tissues, in both the cytoplasm and stroma, was notably elevated (41.88% and 40.63%, respectively) compared to in normal tissues (*p* = 0.008, and *p* = 0.002, respectively). The CXCL-1 expression in the cytoplasm was associated with the tumor status, nodal spread, and distant metastasis. Additionally, a high CXCL-1 level in the stroma was correlated with perineural invasion, the tumor classification, and the TNM stage. Elevated CXCL-1 has been identified as an independent prognostic factor for PC and may serve as a potential therapeutic target and prognostic marker [75]. Zhang et al. reported an association between CXCR-4/CXCL12 and tumor invasion and metastasis. Their study investigated the relationship between the expression of CXCR-4/CXCL12 and vascular endothelial growth factor-C (VEGF-C), Ki-67, matrix metalloproteinase 2 (MMP-2), and β-catenin. The expression of CXCR-4 (CXCL12) was elevated in PC cells (56.7% (86.7%)), adjacent non-cancerous cells (50.0% (85.0%)), and the lymph nodes (53.3% (80.0%)) in comparison to normal controls [76]. CCL-20, a chemotactic cytokine responsible for recruiting inflammatory cells, has been demonstrated to enhance the migration of PC cells. Kimsey et al. demonstrated that increasing the CCL-20 concentration led to a dose-dependent increase in the PC invasion of type IV collagen [77]. Yet, there is a limited understanding of the efficacy of assessing chemokine concentrations in the detection of early-stage PC. There is currently a lack of clinical research investigating chemokines as early biomarkers of the disease.

Several studies have proposed the use of multiple biomarkers or biomarker panels for early diagnosis, as outlined in Table 1. The use of a single tumor marker is reported to have a high probability of false positives and false negatives [78,79]. Park et al. were able to report a sensitivity of 82.5%, a specificity of 92.1%, and an AUC of 0.93 (*p* < 0.01) when using a proteomic multi-marker panel that included LRG1, TTR, and CA19-9, which were 10% higher compared to the sensitivity, specificity, and AUC obtained when using CA19-9 alone [80]. Another study that employed a panel of three biomarkers (CA 19-9, ICAM-1, and osteoprotegerin (OPG)) successfully discriminated healthy patients from those with PDAC, achieving a sensitivity of 88%, a specificity of 90%, and an AUC of 0.93 [81]. In a Korean study, Kim et al. developed a new biomarker combination consisting of apolipoprotein A (ApoA1), CA125, CA19-9, the CEA, ApoA2, and TTR, with a sensitivity, specificity, and area under the curve of 93%, 96%, and 0.993, respectively [82]. Interestingly, all six biomarkers used are part of a pan-diagnostic kit that is commercially available in Korea to diagnose seven cancers: hepatocellular carcinoma, breast cancer, lung cancer, gastric cancer, colon cancer, prostate cancer, and ovarian cancer. In a case-control study conducted by Mellby et al., the differentiation between stages I and II and normal controls yielded a sensitivity and specificity of 94% and 95%, respectively. The biomarker signatures, comprising 29 biomarkers, demonstrated an AUC of 0.96 [83].

*Micro-RNA* (*miRNA*) is single-stranded RNA that was discovered in 1993 and that consists of 19–25 nucleotides [84,85]. Although they are not translated into proteins, *miRNAs*, which are a type of non-coding RNA, play a crucial role in the development and function of the normal human body, influencing processes such as cell division, differentiation, apoptosis, and angiogenesis. *miRNAs* can be classified based on their location (cytoplasmic or nuclear) and length, with small (<200 base-pairs) or long (>200 base-pairs) *miRNAs* [45,86]. *miRNAs* have been associated with tumorigenesis and progression, impacting apoptosis escape, the epithelial–mesenchymal transition (EMT), invasion, and clinical outcomes. The EMT is a phenomenon whereby epithelial cells undergo a transformation, losing their cell-to-cell adhesion and acquiring invasive characteristics akin to mesenchymal cells. This process plays a crucial role in the metastasis of PC [87,88].

The expression of *miRNAs* is influenced by DNA alterations such as deletion, amplification, translocation, and integration during the process of carcinogenesis. Consequently, certain cancers may result in the detection or overexpression of *miRNAs*, making these *miRNAs* potential biomarkers [45]. Various methods, including reverse transcription-quantitative PCR (RT-qPCR), in situ hybridization, next-generation sequencing, and *miRNA* microarrays, can be employed to detect *miRNAs* in blood serum, plasma, cells, and tissues [29,88,89]. In a comprehensive four-stage study utilizing qRT-PCR assays, Zhou et al. identified a six-*miRNA* signature (miR-122-5p, miR-125b-5p, miR-192-5p, miR-193b-3p, miR-221-3p, and miR-27b-3p) that was capable of distinguishing PC patients from normal controls, achieving an AUC of 0.977 (95% CI: 0.894–0.979; sensitivity = 88.7%; and specificity = 89.1%) [90]. Additionally, they reported that miR-125b-5p could serve as an independent biomarker for predicting the survival rates of PC patients.

Serum *miR-25* has been reported to be overexpressed in patients with PDAC. Zhang et al. reported that *miR-25* in pancreatic duct epithelial cells can be maturated in an excessive amount by cigarette smoke condensate (CSCC) [91]. High levels of *miR-25* and *miR-25-3p* suppress PH domain leucine-rich repeat protein phosphatase 2 (PHLPP2), which results in the malignant phenotype of pancreatic cells via the activation of oncogenic AKT-p70S6K signaling. The overexpression of *miR-25-3p* is correlated with a worse prognosis in PC patients [91]. The overexpression *of miR-25* has also been reported in gastric cancer, lung cancer, and cholangiocarcinoma; other studies have suggested that *miR-25* serves as a tumor suppressor in thyroid cancer and colon cancer [92,93,94,95,96]. When *miR-25* was combined with CA19-9 to differentiate PC patients from normal controls, an AUC-ROC of 0.985, a sensitivity of 97.5%, and a specificity of 90.11% were achieved. For the identification of stage I and II tumors, the combination of *miR-25* and CA19-9 accurately detected 40 out of 42 patients (95.24%). These results imply that *miR-25* could potentially function as a novel biomarker for the early detection of PC [97].

Schultz et al. successfully identified two panels of *miRNAs* that are dysregulated in PC [98]. Panel 1 consisted of *miR-145*, *miR-150*, *miR-223*, and *miR-636*, and Panel 2 consisted of *miR-26b*, *miR-34a*, *miR-122, miR-126*, *miR-145*, *miR-150*, *miR-223*, *miR-505, miR-636*, and *miR-885.5p*. These *miRNA* panels were capable of distinguishing PC patients from healthy subjects. Using Panel 1, the study attained an AUC of 0.86 (95% CI: 0.82–0.90), a sensitivity of 0.85 (95% CI: 0.79–0.90), and a specificity of 0.64 (95% CI: 0.57–0.71). Panel 2 yielded an AUC of 0.93 (95% CI: 0.90–0.96), a sensitivity of 0.85 (95% CI: 0.79–0.90), and a specificity of 0.85 (95% CI: 0.80–0.85). Interestingly, when combined with CA19–9, both panels were able to detect PC stages IA–IIB with the following performance: Panel 1 with an AUC of 0.83 (95% CI: 0.76–0.90) and Panel 2 with an AUC of 0.91 (95% CI: 0.86–0.95). In a similar investigation conducted by Johansen et al., four panels were employed, namely Panel I (comprising seven *miRNAs*), Panel II (comprising nine *miRNAs*), Panel III (comprising five *miRNAs*), and Panel IV (comprising twelve *miRNAs*). The patients diagnosed with PC in Panels I and II were contrasted with a combined group of individuals with chronic pancreatitis and those who were healthy. Conversely, the patients with PC in Panels III and IV were compared specifically to healthy participants (refer to Table 2). Panels I and III were designed to be robust to technical variation, and Panels II and IV included all the significant *miRNAs* from a multivariate model, thus representing the upper limit in terms of training [99]. The best panel for discriminating stages I and II PC from healthy subjects was Panel II combined with serum CA19-9, which exhibited a sensitivity of 0.77 (0.69–0.84), a specificity of 0.94 (0.90–0.96), and an AUC of 0.93 (0.90–0.96). It is noteworthy that the aforementioned studies did not share any *miRNAs* in their panels, except for *miR-25*.

In addition to their presence in serum and pancreatic tissue samples, *miRNAs* are also present in feces, urine, and saliva. *miR-143, miR-223*, and *miR-30* can be detected in urine even in stage I cancer. The joint utilization of *miR-143* and *miR-30* exhibited a sensitivity and specificity of 83.3% and 96.2%, respectively, with an AUC of 0.92 [100,101]. Assessing the *miR-1246* and *miR-4644* levels in saliva has been studied to differentiate PC patients from healthy controls, yielding AUC values for the ROC curves of 0.814 (*p* = 0.008) and 0.763 (*p* = 0.026), respectively. Combining *miR-1246* and *miR-4644* increased the AUC to 0.833 (*p* = 0.005) [102]. Salivary *miRNAs* were reported to be stable due to the protection provided by exosomes. In the stool samples of PC patients, *miR-21* and *miR-155* exhibited overexpression (*p* = 0.0049 and *p* = 0.0112, respectively), while *miR-216* showed lower expression levels (*p* = 0.0002). The combination of *miR-21*, *miR-155*, and *miR-216* for PC screening demonstrated a sensitivity of 83.3%, a specificity of 83.3%, and an AUC of 0.866 (95% CI: 0.7722–0.9612) [103].

### 3.2. Circulating DNA

*Circulating tumor DNA* (*ctDNA*) was first described in 1948, and it has been postulated that the DNA release via the necrosis, apoptosis, and lysis of circulating tumor cells (CTCs) and micro-metastasis contributes to the presence of *ctDNA* [104,105,106]. *ctDNA* comprises 170–181 base-pairs and is present in body fluids at very low concentrations, ranging from 1 to 100 ng/mL, depending on the type and tumor burden [79,106]. Due to its low concentration in body fluids, detecting *ctDNA* requires methods with a high analytical sensitivity and specificity. The methods used to detect *ctDNA* include real-time PCR, automatic sequencing, mass spectrometry genotyping, next-generation sequencing (NGS), and digital PCR platforms (such as digital droplet PCR (ddPCR)). The sensitivity of these methods greatly varies, ranging between 0.01% and 15% [107,108,109,110].

*ctDNA* levels have been reported to be elevated in patients with PC. In a study by Shapiro et al., *ctDNA* levels as low as 25 ng/mL were detected by utilizing radioimmunoassay DNA quantification, with DNA levels exceeding 100 ng/mL being considered the upper normal limit [111]. The *KRAS* gene has received significant attention in terms of *ctDNA* mutations because it is highly mutated in PC [108]. An assessment of samples from 26 PC patients for 54 genes revealed that *KRAS*, *TP53*, *APC*, *FBXW7*, and *SMAD4* may be potential markers for detecting pancreatic ductal adenocarcinoma (PDAC) [112]. A *ctDNA KRAS* mutation for the diagnosis of PDAC was reported to have a sensitivity of 47% and a specificity of 87%, and when combined with CA19-9, it had a sensitivity of 98% and a specificity of 77% [113]. On the contrary, Cohen et al. reported that CA19-9 outperformed *ctDNA* in the detection of stages I and II PDAC [114]. The results of studies on *ctDNA* have been varied. In a study of 26 cancer patients utilizing next-generation sequencing (NGS) technology, *KRAS*, *TP53*, *APC*, *FBXW7*, and *SMAD4* mutations were found in 90% of the matched tumor biopsies. The diagnostic accuracy was reported to be 97.7%, with an average sensitivity of 92.3% and a specificity of 100% across all five investigated genes [115]. Conversely, Pishavian et al. reported an overall concordance of only 25% between blood and tissue samples using NGS assays, and *KRAS* mutations were detected in only 29% of the blood samples compared to 87% of the tumor tissue biopsies [116]. Similarly, in another study evaluating the correspondence of *KRAS* mutations in PC tissue and *ctDNA*, researchers reported that *KRAS* mutations were detected in 70% of neoplastic tissue samples, but none were found in *ctDNA* samples [117].

Currently, the use of *ctDNA* as a diagnostic tool is limited due to the low amount of detectable *ctDNA* in the early stage of the disease [118]. However, *ctDNA* has shown a correlation with the tumor burden and holds promise as a tool for predicting the treatment response and for monitoring in advanced cases [119]. Chen et al. found a correlation between *KRAS-mutant ctDNA* and both the time to progression and the overall survival. The detection rates in patients with non-elevated CA19-9 were 93.7% and 86.4%, respectively. *KRAS* mutations were also able to correctly predict 80% of the patient response to treatment [120]. Patients with *KRAS-mutant ctDNA* were reported to have 6.1 months of disease-free survival in comparison to 16.1 months in patients who had no such mutation, with overall survival times of 13.3 and 27.6 months, respectively (*p* < 0.001) [121]. Similarly, a recent study using digital droplet PCR (ddPCR) reported that *KRAS-mutated ctDNA* was associated with a poorer prognosis of 170 days versus 489 days; notably, the presence of a *KRAS* mutation in tissue DNA did not show a similar association with survival rates [122]. A specific subtype of *KRAS* mutation, *p.G12V*, was linked to a shorter survival time compared to *p.G12D*, *p.G12R*, or wild-type variants [122]. Serial plasma testing of *KRAS-mutant ctDNA* in advanced PDAC patients undergoing chemotherapy appears to provide more effective monitoring than CA19-9 [123]. The longitudinal monitoring of *ctDNA* has been reported to predict a patient’s response to therapy and disease progression around 5 months earlier than standard radiological imaging and CA19-9 [124,125].

The application of *ctDNA* is currently restricted due to the inconsistent concordance between tissue biopsies and liquid biopsies, which range widely from 48% to 100% [121]. Additionally, the lack of standardized protocols, variations in the reliability of the *ctDNA* detection methods across studies, and limited validation studies further contribute to the limitations [108]. Moreover, given that mutations are not exclusive to PC and can be observed in other tumor entities, there are challenges to achieving high diagnostic sensitivity and specificity [79].

### 3.3. Circulating Tumor Cells (CTCs)

Intact cells released by tumors, known as circulating tumor cells, can be identified in the bloodstream [126]. After shedding, the circulating tumor cells can disseminate through blood vessels and invade local tissue stoma [127,128,129]. It has been reported that CTCs can be detected before metastasis [130]. CTCs were reported to be present in whole blood at a ratio of around 1 for every 107 leukocytes per mL, with a half-life estimated at around 1–2.4 h [131]. The identification of CTCs includes the process of CD45 depletion to remove leukocytes; then, the enrichment of CTCs is performed via size-based filtration or through the use of epithelial cell adhesion molecules (EPCAMs). Actual CTC recognition involves examining the cell morphology and measuring the expression of particular gene markers or proteins via the immunofluorescence of molecules specific to CTCs [79,131]. Another CTC detection method utilizes genomic, transcriptomic and proteomic approaches; one of the most widely used is the FDA-approved Cell Search^®^ [131]. Developing methods for detecting CTCs is challenging, as there are only a low number of captured CTCs [132]. Furthermore, PC has been reported to have a lower detection rate in comparison to other tumors [108].

Numerous studies have indicated that circulating tumor cells (CTCs) may exhibit a sufficient sensitivity for the detection of stage I and II PC. Kulemann et al. detected CTCs in early-stage IIA and IIB tumors in 8 out of 10 patients (80%) using immunofluorescence for an epithelial-to-mesenchymal transition (EMT) marker and an epithelial antigen cytokeratin (CK); in contrast, CTCs were absent in all 10 control patients, with a *p*-value of less than 0.001 [133]. Similarly, Xu et al. were able to detect CTCs in 90% of PC patients using the negative enrichment (NE), immunofluorescence, and in situ hybridization (FISH) of chromosome 8 (NE-iFISH); when combined with CA19-9, the diagnostic rate was reported to reach 97.5%, with a rate of 75% for benign disease and 73% for early-stage PC [134]. Furthermore, Rhim et al. reported capturing CTCs in 33% of patients with cystic lesions without a clinical diagnosis of cancer (Sendai criteria), 73% of patients with PDAC, and no detection in patients without cysts or cancer [135]. Although promising, the use of CTCs as an early biomarker is not yet suitable for clinical settings and still requires studies involving a higher number of samples.

## 4. Artificial Intelligence (AI)

The application of artificial intelligence (AI) can mitigate the subjectivity of doctors and address inconsistent diagnoses arising from variations in training, experience, and professional attributes [14]. Diagnosing PC demands the expertise of professionals to conduct intricate analyses involving vast amounts of data, encompassing imaging, pathological slices, and biomarkers. Artificial neural networks (ANNs), which are nonparametric machine-learning models that emulate human brain processing, consist of processing elements called neurodes arranged in layers to simulate the hierarchical activation of neurons in the brain [136]. In one study, ANN models were reported to predict the 7-month survival of PDAC patients, with or without resection, and these models achieved a 91% sensitivity and a 38% specificity [136]. Yang et al. demonstrated that, when applied to 913 serum specimens for the analysis of CA19-9, CA-125, and CEA, the ANN outperformed each serum tumor marker alone and a logistic regression model. The sample size of 913 was randomly divided into a training group (*n* = 658) and a test group (*n* = 225), revealing an AUC for the ANN of 0.905 (95% confidence interval [CI]: 0.868–0.942) compared to 0.812 (95% CI: 0.762–0.863) for the logistic regression model [137]. Another team employed a penalized algorithm to create a PC diagnostic model with 29 *miRNA* markers in 63 PC patients and 63 controls, validated with 25 PC samples and 81 intrahepatic cholangiocarcinoma patients’ serum samples. The algorithm-based diagnostic model demonstrated a sensitivity of 96% and a specificity of 90%, surpassing CA19-9 with sensitivity and specificity levels that were 1.5 and 2 times higher, respectively [138].

## 5. Nanoparticles (NPs)

NPs, with sizes ranging from 1 to 100 nm, have emerged as a category of materials holding promise for applications in in vivo imaging and biological diagnostics. As contrast agents, NPs are ideally characterized by an easy dispersibility; a stability unaffected by factors such as the polarity, ionic strength, pH, or temperature; programmed clearance mechanisms; and sensitivity and selectivity for the target (e.g., antigen, cell, tissue) [139]. Rosenberger et al. introduced a novel NP developed as an MRI contrast agent for PC. This NP was composed of recombinant human serum albumin (rHSA) incorporating iron oxide (maghemite, γ-Fe_2_O_3_) with a strong affinity for galectin-1. Galectin-1 was selected as the target receptor due to its overexpression in PC and its precursor lesions, but not in normal pancreatic tissue or pancreatitis [140]. Another study by Luo et al. reported the use of hyaluronic acid (HA)-mediated multifunctional Fe_3_O_4_ NPs to target PC cancer cells via CD44 on the cytoplasmic membrane, which has a high affinity for HA. HA-Fe_3_O_4_ NPs were shown to be efficient T2-weighted magnetic resonance imaging (MRI) contrast agents [141]. The application of NPs to enhance the response of antigen–antibody sensing processes, utilizing SiO_2_ nanoparticle-labeled secondary antibodies, was described in 2010 by Zhuo et al. The results indicated that the CA19-9 detection threshold was 100 times lower than that of the traditional ELISA method, potentially enabling the detection of early-stage PC changes in CA19-9 levels [142]. Another study utilized multiwalled carbon nanotube (MWCNT) paper for the detection of CA19-9, demonstrating the ability to detect a wide range of CA19-9 concentrations (0 U/mL to at least 1000 U/mL). This method involved adding CA19-9 antibodies to the surface of MWCNTs deposited on microporous filter paper, and the resistance of the biosensor element was linear to the concentration of CA19-9 [143].

## 6. Conclusions

With PC presenting with a poor prognosis, especially due to late-stage detection, the imperative for early diagnostic tools is evident. Establishing an accurate and targeted screening model for PC is crucial. Optimal biomarkers should efficiently distinguish between healthy individuals and patients, enable early detection, allow for ease of measurability, be cost-effective, and yield reproducible results.

Recent investigations into novel biomarkers have shown promise, but they require further validation with larger sample sizes and standardized measurement methods. The cost-effectiveness and practicality are crucial considerations for the widespread clinical application of these biomarkers. The authors emphasize the importance of focused research on realistic and applicable early detection methods and models for clinical use.

Identifying a singular biomarker for the early detection of PC poses significant challenges. The authors concur that the most effective approach currently involves integrating various factors, such as environmental factors, risk scores, genetics, and biomarkers. Those individuals who meet the risk threshold criteria from a clinical diagnostic model should undergo follow-up in surveillance programs, incorporating a combination of biomarkers. A comprehensive cancer detection tool employing established biomarkers (ApoA1, CA125, CA19-9, CEA, ApoA2, and TTR) has demonstrated promise in predicting multiple cancers at a reasonable cost. The pan-diagnostic tool, which is accessible in Korea, employs several renowned biomarkers and has been documented to anticipate various cancers at a price of USD 300, with a reported AUC of 0.993 for the detection of PC [82]. Enhancements to these models could aid in identifying high-risk individuals earlier, emphasizing the need to screening high-risk groups at an early stage.

## Figures and Tables

**Table 1 biomolecules-14-00364-t001:** Biomarkers and combination biomarkers for the detection of PC.

Study Reference	Study Method	Material	Sample Size	Biomarkers	Sensitivity	Specificity	AUC
Joergensen [63]	Prospective case control	Blood	(PDAC/normal) 51/52	TIMP-1	47.1%	69.2%	0.64
Chen [66]	Prospective case control	Serum	(PDAC/normal) 67/62	TTR	90.5%	47.6%	0.75
Mohamed [67]	Cohort	Serum	(PDAC/non cancer) 50/27	ICAM-1	82%	82.6%	0.85
Park [80]	Retrospective cohort	Plasma	(PDAC/non PDAC) 401/607	LRG1, TTR, and CA19-9	82.5%	92.1%	0.93
Brand [81]	Cohort	Serum	(PDAC/healthy) 160/107	ICAM-1, OPG, and CA19-9	88%	90%	0.93
Kim [82]	Cohort	Blood	(PDAC/healthy) 180/573	ApoA1, CA125, CA19-9, CEA, ApoA2, and TTR	93%	96%	0.993
Mellby [83]	Case control	Blood	(PDAC/healthy) 443/888	Panel of 29 biomarkers	94%	95%	0.96

**Table 2 biomolecules-14-00364-t002:** The four investigated diagnostic *microRNA* panels: Panels I and II contrasted patients with PC against those with chronic pancreatitis and healthy individuals, while Panels III and IV solely compared PC patients to healthy individuals [99].

Panel I	Panel II	Panel III	Panel IV
*miR-16*	*miR-16*	*miR-16*	*miR-16*
*miR-27a*	*miR-24*	*miR-27a*	*miR-18.a*
*miR-30a.5p*	*miR-27.a*	*miR-25*	*miR-24*
*miR-323.3p*	*miR-30a.5p*	*miR-29c*	*miR-27a*
*miR-20a*	*miR323.3p*	*miR-483.5p*	*miR30a.5p*
*miR-29c*	*miR-20a*		*miR-323.3p*
*miR-483.5p*	*miR-25*		*miR-20a*
	*miR-29c*		*miR-25*
	*miR-483.5p*		*miR-29c*
			*miR-191*
			*miR-345*
			*miR-483.5p*

## Data Availability

Not applicable.

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
