# Peer review of "A Quest for Survival: A Review of the Early Biomarkers of Pancreatic Cancer and the Most Effective Approaches at Present"

_biomolecules, 2024, doi:10.3390/biom14030364_

Round 1

Reviewer 1 Report (Previous Reviewer 2)

Comments and Suggestions for Authors

Thank you. All our concerns have been addressed. We have no more critiques.

Author Response

Dear reviewers,

Thank you very much for taking the time to review this manuscript.

Comments 1: Thank you. All our concerns have been addressed. We have no more critiques.

Response 1: Thank you for your valuable feedback and thoughtful considerations. We are pleased that your insights and comments have contributed to the enhancement of our manuscript.

Sincerely,

M. Begawan Bestari, MD, MHS, PhD, FASGE, FACG

Reviewer 2 Report (Previous Reviewer 1)

Comments and Suggestions for Authors
The review entitled “Early Biomarkers for Pancreatic Cancer: A Quest for Survival, Review on the Most Effective Approaches at Present” includes two basic section: characteristics of pancreatic cancer and the review of the significance of potential biomarkers for this deadly malignancy.  All comments from my last review were taken into account when preparing the review for resubmission. However, I only have one comment regarding the conclusion. There is still no specific information as to which of the potential markers described in the paper could, in the author's opinion, the most important indicator in the screening and diagnosis of PC.

Comments on the Quality of English Language

The review entitled “Early Biomarkers for Pancreatic Cancer: A Quest for Survival, Review on the Most Effective Approaches at Present” includes two basic section: characteristics of pancreatic cancer and the review of the significance of potential biomarkers for this deadly malignancy.  All comments from my last review were taken into account when preparing the review for resubmission. However, I only have one comment regarding the conclusion. There is still no specific information as to which of the potential markers described in the paper could, in the author's opinion, the most important indicator in the screening and diagnosis of PC.

Author Response

Dear reviewers,

Thank you very much for taking the time to review this manuscript. Please find the detailed responses below and the corresponding revisions.

Comments 1: Quality of English Language (x) Minor editing of English language required.

Response 1: The manuscript, along with its initial draft, has undergone proofreading by language editing services provided by MDPI. It has been meticulously reviewed for accurate grammar usage and proper incorporation of common technical terms. Additionally, we have included the English editing certificate with this submission

Comments 2: All comments from my last review were taken into account when preparing the review for resubmission. However, I only have one comment regarding the conclusion. There is still no specific information as to which of the potential markers described in the paper could, in the author's opinion, the most important indicator in the screening and diagnosis of PC.

Response 2: We have, accordingly, revised the conclusion to emphasize this point.
In conclusion section paragraph 3 we have added specific information as to which of the potential markers described in the paper could, in the author's opinion, the most important indicator in the screening and diagnosis of PC. (In the revised mauscript changes are highlighted in yellow)

“Identifying a singular biomarker for the early detection of PC poses significant challenges. The authors concur that the most effective approach currently involves integrating various factors such as environmental, risk scores, genetics, and biomarkers. Those individuals meeting the risk threshold criteria from a clinical diagnostic model should undergo follow-up in surveillance programs, incorporating a combination of biomarkers. Comprehensive cancer detection tool employing established biomarkers (ApoA1, CA125, CA19-9, CEA, ApoA2, and TTR) has demonstrated promise in predicting multiple cancers at a reasonable cost. The pan-diagnostic tool, which is accessible in Korea, employs several renowned biomarkers and has been documented to anticipate various cancers at a price of USD 300., with a reported AUC of 0.993 for the detection of PC.”

Thank you for your feedback and consideration. I eagerly anticipate your response.

Sincerely,

M. Begawan Bestari, MD, MHS, PhD, FASGE, FACG

This manuscript is a resubmission of an earlier submission. The following is a list of the peer review reports and author responses from that submission.

Round 1

Reviewer 1 Report

Comments and Suggestions for Authors

The review by Bestari entitled “Early Biomarkers for Pancreatic Cancer: A Quest for Survival, Review on the Most Effective Approaches at Present” includes two basic section: characteristics of pancreatic cancer and potential biomarkers.

This manuscript fulfils the expectations of a summary of the literature on the potential usefulness of selected biomarkers in pancreatic cancer diagnosis and therapy. Nevertheless, the manuscript contains severe errors and needs to be heavily revised.

1.       Introduction.

-          Line 31 – at later stage??? – please check

2.       Current biomarkers

-          please describe which non-cancer diseases cause an increase in CEA concentrations

-          Line 107 – add units

3.       Novels biomarkers

-          Line 119 – correct the sentence concerning TIMP-1. Please describe the role of MMPs as biomarkers for PC, as well as that the main role of TIMP-1 is regulation of MMP-9 activity. Moreover, short characteristic concerning other potential biomarkers such as selected chemokines or claudins should be also presented.

4.       Section 3.2. should be placed into 3.1

5.       Table 1 – please add which methods were used and which material was used for the analyses.

6.       Line 180-184 – please describe the method used for this study

7.       Table 2 is unclear – differentiation groups should be moved from the title of the table 2 to the body of the table

8.       Screening Feas… – the information concerning PC content is similar to introduction section, so please move section 4 in the introduction.

9.       Conclusion - must be completely modified because it does not summarize the most important conclusions from the work, i.e. which biomarkers have the best potential for use in the diagnosis of pancreatic cancer

Minor issues (a selection):

1.       Abbreviations are not presented in a correct way (line 144, 125)

2.       Please use PC abbreviation in a whole paper

3.       Term: “ were able” is used too often and in wrong meaning.

4.       Repetitions in words in one sentence – line 199, line 200

5.       Rephrase the sentence, it is unclear: lines 353-356; 61-61; 129-131,203-207

6.       Presentation of statistical significance value – “p” or “P” – check the Authors instruction.

Extensive editing of English language required!

Comments on the Quality of English Language

Extensive editing of English language required

Reviewer 2 Report

Comments and Suggestions for Authors

There is a lot of literature reviewing biomarkers in pancreatic cancer, including notable references such as 10.1155/2018/5389820, 10.3390/cells13010003, and 10.1186/s12943-020-01245-y, et al. Various perspectives on biomarker analysis, such as exosomes, non-coding RNAs (ncRNAs), and liquid biopsy, have been explored in these studies.

1.Novelty of the Paper:

I am intrigued by the paper's title, which suggests a quest for survival and the identification of the most effective approaches currently available. However, upon examination, the entire paper does not consistently revolve around this central theme.

2.Definition of "Most Effective Approaches":

I am interested in understanding how the paper defines "most effective approaches" and the criteria used for study inclusion and exclusion. Additionally, clarification on how the effectiveness rate is calculated would be beneficial. Some biomarkers mentioned in the review, like TIMP-1, are reported to have low sensitivity (47.1%), specificity (69.2%), and an AUC of 0.64 in diagnosing pancreatic cancer. Understanding the basis for these assessments would contribute to a more comprehensive evaluation of the discussed biomarkers.

3.New Technology:

It would be valuable to incorporate information on emerging technologies, such as Artificial Intelligence (AI), and a spectrum of circulating tumor-derived analytes (including proteins, autoantibodies, metabolites, DNA, non-coding RNAs, and extracellular vesicles). This addition could provide insights into the cutting-edge advancements shaping the field.